# Generalized Maximum Complex Correntropy Augmented Adaptive IIR Filtering

**DOI:** 10.3390/e24071008

**Published:** 2022-07-21

**Authors:** Haotian Zheng, Guobing Qian

**Affiliations:** College of Electronic and Information Engineering, Chongqing Key Laboratory of Nonlinear Circuits and Intelligent Information Processing, Southwest University, Chongqing 400715, China; zhenghaotian151@foxmail.com

**Keywords:** GMCCC, complex, augmented IIR, non-Gaussian noise, system identification

## Abstract

Augmented IIR filter adaptive algorithms have been considered in many studies, which are suitable for proper and improper complex-valued signals. However, lots of augmented IIR filter adaptive algorithms are developed under the mean square error (MSE) criterion. It is an ideal optimality criterion under Gaussian noises but fails to model the behavior of non-Gaussian noise found in practice. Complex correntropy has shown robustness under non-Gaussian noises in the design of adaptive filters as a similarity measure for the complex random variables. In this paper, we propose a new augmented IIR filter adaptive algorithm based on the generalized maximum complex correntropy criterion (GMCCC-AIIR), which employs the complex generalized Gaussian density function as the kernel function. Stability analysis provides the bound of learning rate. Simulation results verify its superiority.

## 1. Introduction

Complex-valued adaptive filtering algorithm has a wide range of engineering applications in radio systems [1], system identification [2], environment signal processing [3], and other fields. Generally speaking, complex-valued adaptive filtering algorithm is an extension of the real-valued adaptive filtering algorithm. When the complex signal is second-order circular (or proper) [3], the performance of adaptive filter is optimal. For second-order circular signals, the covariance matrix $C_{xx}=E\left\{x\left(n\right)x^{H}\left(n\right)\right\}$ is second-order statistics. A complex-valued random variable is second-order circular if its first and second-order statistics are rotation-invariant in the complex plane, but in most cases, complex signals are noncircular (or improper) [3].

In order to suit both proper and improper complex-valued signals, augmented complex statistics are proposed. There are quantities of adaptive filtering algorithms based on augmented complex statistics, such as augmented complex least mean square (ACLMS) [4], augmented complex adaptive infinite impulse response (IIR) algorithm (ACA-IIR) [5], diffusion augmented complex adaptive IIR algorithm (DACA-IIR) [6], and incremental augmented complex adaptive IIR algorithm (IACA-IIR) [7]. These adaptive filtering algorithms are based on the mean square error (MSE) criterion, which is mathematically tractable, computationally simple, and optimal under Gaussian assumptions [8]. However, the MSE-based algorithm may perform poorly or encounter instability problems when the signal is disturbed by non-Gaussian noise [9,10]. From a statistical point of view, the mean square error is not sufficient to capture all possible information in a non-Gaussian signal. In practical applications, non-Gaussian noise is common. For example, some sources of non-Gaussian impulse noise are non-synchronization in digital recording, motor ignition noise in internal combustion engines, and lightning spikes in natural phenomena [11,12].

Entropy generally describes a measure of uncertainty of a real random variable, and as a means of a functional analysis method, entropy of a signal can define the noise without using a threshold criterion [13]. Correntropy is an extension of entropy, which is a quantity of how similar two random complex variables are in a neighborhood of the joint space controlled by the kernel bandwidth. Compared with the MSE-based algorithms, correntropy algorithm is superior. On the whole, the correntropy uses the Gaussian function as the kernel function [14,15], because it is smooth and strictly positive definite. However, the Gaussian kernel is not always an appropriate choice. Recently, He et al. [16] and Chen et al. [17] extended it to more general cases and proposed the generalized maximum correntropy criterion (GMCC) algorithm, which has strong generality and flexibility, and Qian et al. [18] proposed a GMCCC algorithm based on the generalized maximum complex correntropy criterion, which uses the complex generalized Gaussian density(CGGD) function as a kernel of the complex correntropy. It succeeds in the excellent characteristics of GMCC and can deal with complex signals at the same time. These correntropy algorithms are finite impulse response (FIR) wide linear adaptive filtering algorithms, but when FIR filters need to use a large number of coefficients to obtain satisfactory filtering performance, FIR wide linear models may not always be appropriate.

Unlike the FIR counterpart, the memory depth of an IIR filter is independent of the filter order and the number of coefficients [19]. Alternatively, an IIR filter generally requires considerably fewer coefficients than the corresponding FIR filter to achieve a certain level of performance. Thus, the IIR adaptive filters are suitable for systems with memory, such as autoregressive moving average (ARMA) models. Navarro-Moreno et al. [20] developed an ARMA widely linear model with fixed coefficients. To derive a recursive algorithm for augmented complex adaptive IIR filtering, Took et al. [7] proposed the ACA-IIR to learn the parameters of a widely linear ARMA model.

Based on the generalized maximum complex correntropy criterion (GMCCC) and widely linear ARMA model, we propose a GMCCC algorithm variant, namely the GMCCC augmented adaptive IIR filtering algorithm (GMCCC-AIIR). We show that the GMCCC-AIIR is very flexible, with ACA-IIR, GMCCC, and ACLMS as its special cases. Stability analysis shows that GMCCC-AIIR always converges when the step-size satisfies the theoretical bound. Simulation results demonstrate the superiority of the GMCCC-AIIR algorithm.

The organization of the paper is as follows: Section 2 introduces and describes the augmented IIR system. Section 3 defines the generalized complex correntropy, derives the GMCCC-AIIR algorithm, and introduces a reduced-complexity version of the proposed algorithm. Section 4 provides the analysis on the bounds of the step-size for convergence. The superiority of the GMCCC-AIIR algorithm is verified by simulations in Section 4, and the conclusion is drawn in Section 5.

## 2. Augmented IIR System

The signals used in communications are usually complex circular, whereas the class of signals made complex by convenience of representation become more general, and such signals are often noncircular. For the stochastic modeling of this kind of signal, Picinbono et al. introduce a widely linear moving average (MA) model, which is given by [21]: (1)$$y\left(n\right)=\sum_{m=0}^{N}b_{m}x\left(n-m\right)+\sum_{m=0}^{N}h_{m}x^{*}\left(n-m\right)$$
where $b_{m}$ and $h_{m}$ are filter coefficients. Based on this widely linear model, an ACLMS algorithm was proposed [22].

Since the FIR generalized linear model is not always an optimal choice, Moreno et al. introduce a fixed coefficient ARMA generalized linear model [20].
(2)$$\begin{array}{ccc} y\left(n\right) & = & \sum_{m=1}^{p}a_{m}y\left(n-m\right)+\sum_{m=0}^{q}b_{m}x\left(n-m\right)+\sum_{m=1}^{p}g_{m}y^{*}\left(n-m\right)\\ & \; & +\sum_{m=0}^{q}h_{m}x^{*}\left(n-m\right) \end{array}$$
where $a_{m}$, $g_{m}$ are the coefficients of feedback and its conjugation, *p* and *q* are the orders of the AR and MA parts, respectively. The model provides a theoretical basis for the proposed recursive algorithm for training adaptive IIR filters.

To introduce a recursive algorithm of augmented complex adaptive IIR filter, Took et al. [5] give the output of the widely linear IIR filter in the following form:(3)$$\begin{array}{ccc} y\left(n\right) & = & \sum_{m=1}^{M}a_{m}\left(n\right)y\left(n-m\right)+\sum_{m=0}^{N}b_{m}\left(n\right)x\left(n-m\right)+\sum_{m=1}^{M}g_{m}\left(n\right)y^{*}\left(n-m\right)\\ & \; & +\sum_{m=0}^{N}h_{m}\left(n\right)x^{*}\left(n-m\right) \end{array}$$
where *M* is the order of the feedback and *N* is the length of the input. This model can be simplified as follows: (4)$$y\left(n\right)=w^{T}\left(n\right)z\left(n\right)$$
where: (5)$$w\left(n\right)={\left[a_{1}\left(n\right),\ldots ,a_{M}\left(n\right),g_{1}\left(n\right),\ldots ,g_{m}\left(n\right),b_{0}\left(n\right),\ldots ,b_{M}\left(n\right),h_{0}\left(n\right),\ldots ,h_{M}\left(n\right)\right]}^{T}$$
(6)$$z\left(n\right)={\left[y^{T}\left(n\right),x^{T}\left(n\right)\right]}^{T}$$

## 3. Generalized Complex Correntropy and GMCCC-AIIR Algorithm

### 3.1. Generalized Complex Correntropy

For two complex variables $C_{1}=X+jY$ and $C_{2}=Z+jS$, complex Correntropy is defined as [23]:(7)$$\left(C_{1},C_{2}\right)=E\left[\kappa \left(C_{1}-C_{2}\right)\right]$$
where X,Y,Z,S are real variables, $\kappa (C_{1}-C_{2})$ is the kernel function.

For the Gaussian kernel in the complex field [23], the kernel function can be expressed as: (8)$$\kappa \left(C_{1}-C_{2}\right)=G^{C}\left(C_{1}-C_{2}\right)=\frac{1}{2\pi \sigma^{2}}\exp\left(-\frac{\left(C_{1}-C_{2}\right){\left(C_{1}-C_{2}\right)}^{*}}{2\sigma^{2}}\right)$$
where σ is the kernel width.

In this paper, we employ a CGGD function as the kernel function, and it’s corresponding correntropy is named generalized complex correntropy [18]:(9)$$\begin{array}{ccc} \kappa \left(C_{1}-C_{2}\right) & = & G_{\alpha ,\beta }^{L}\left(C_{1}-C_{2}\right)\\ \; & = & \frac{\alpha }{\pi \beta \Gamma (1/\alpha )}\exp\left\{-\frac{{\left[\left(C_{1}-C_{2}\right){\left(C_{1}-C_{2}\right)}^{*}\right]}^{\alpha }}{\beta^{\alpha }}\right\}\\ \; & = & \gamma_{\alpha ,\beta }\exp\left\{-\lambda {\left[\left(C_{1}-C_{2}\right){\left(C_{1}-C_{2}\right)}^{*}\right]}^{\alpha }\right\} \end{array}$$
where α is the shape parameter, $\beta =2\sigma^{2}\Gamma \left(1/\alpha \right)/\Gamma \left(2/\alpha \right)$ is the kernel width, $\lambda =1/\;\beta^{\alpha }$, $\gamma_{\alpha ,\beta }\;=\;\frac{\alpha }{\pi \beta \Gamma \left(1/\alpha \right)}$.

In this way, generalized complex correntropy can be written as: (10)$$V_{\alpha ,\beta }^{C}\left(C_{1},C_{2}\right)=E\left[G_{\alpha ,\beta }^{C}\left(C_{1}-C_{2}\right)\right]$$

The samples ${\left\{\left(c_{1i},c_{2i}\right)\right\}}_{i=1}^{N}$ are finite in reality, so we estimate the generalized complex correntropy by sample mean.
(11)$$\begin{array}{ccc} \hat{V_{\alpha ,\beta }^{C}}\left(C_{1},C_{2}\right) & = & \frac{1}{N}\sum_{i=1}^{N}G_{\alpha ,\beta }^{C}\left(c_{1i}-c_{2i}\right)\\ \; & = & \frac{1}{N}\sum_{i=1}^{N}G_{\alpha ,\beta }^{C}\left(e_{i}\right) \end{array}$$
where $e_{i}=\;c_{1i}-c_{2i}$.

Instead of the correntropy in data analysis, the correntropic loss is often used, so we define the generalized complex correntropic loss as:(12)$$J_{GC-loss}^{C}\left(C_{1},C_{2}\right)=G_{\alpha ,\beta }^{C}\left(0\right)-V_{\alpha ,\beta }^{C}\left(C_{1},C_{2}\right)$$

Then, when the sample is finite, the generalized complex correntropic can be expressed as: (13)$$\begin{array}{ccc} \hat{J_{GC-loss}^{C}}\left(C_{1},C_{2}\right) & = & \gamma_{\alpha ,\beta }-\frac{1}{N}\sum_{i=1}^{N}G_{\alpha ,\beta }^{C}\left(c_{1i}-c_{2i}\right)\\ \; & = & \gamma_{\alpha ,\beta }-\frac{1}{N}\sum_{i=1}^{N}G_{\alpha ,\beta }^{C}\left(e_{i}\right) \end{array}$$

There are some properties of generalized complex correntropy [18].

**Property** **1.**
*$\;V_{\alpha ,\beta }^{C}\left(C_{1},C_{2}\right)$ is symmetric, i.e., $V_{\alpha ,\beta }^{C}\left(C_{1},C_{2}\right)\;=\;V_{\alpha ,\beta }^{C}\left(C_{2},C_{1}\right)$*


**Property** **2.**
*$\;V_{\alpha ,\beta }^{C}\left(C_{1},C_{2}\right)$ is bounded with $0\leq \;V_{\alpha ,\beta }^{C}\left(C_{1},C_{2}\right)\leq \gamma_{\alpha ,\beta }$ and achieves its maximum when $C_{1}=C_{2}$*


We can get $\;J_{GC-loss}^{C}\left(C_{1},C_{2}\right)$, it is symmetric and achieves its minimum when $C_{1}\;=\;C_{2}$ on the basis of Properties 1 and 2.

**Property** **3.**
*Given ${e=\left[e_{1},e_{2}\ldots e_{N}\right]}^{T}$, the following conclusions about $\hat{J_{GC-loss}^{C}}$ are true:*

*When α≥1/2,$\;\hat{J_{GC-loss}^{C}}$ is convex at any e with $\left|e_{n}\right|\leq {\left[2a-1/2a\lambda \right]}^{1/2a}$;*

*When 0<α<1/2, $\hat{J_{GC-loss}^{C}}$ is non-convex at any e with $\left|e_{n}\right|\;\neq \;0$.*



### 3.2. GMCCC-AIIR Algorithm

Based on properties of generalized complex correntropy, we define the cost function of the GMCCC-AIIR algorithm as: (14)$$\begin{array}{ccc} J_{GC-loss}^{C} & = & G_{\alpha ,\beta }^{C}\left(0\right)-E\left[G_{\alpha ,\beta }^{C}\left(e\left(n\right)\right)\right]\\ \; & = & \gamma_{\alpha ,\beta }\left\{1-E\left[\exp\left[-\lambda {\left(e\left(n\right)e^{*}\left(n\right)\right)}^{\alpha }\right]\right]\right\} \end{array}$$
where $e\left(n\right)=d\left(n\right)-y\left(n\right)$.

We can infer from (4) that $e\left(n\right)=d\left(n\right)-w^{T}\left(n\right)z\left(n\right)$, so $\;e^{*}\left(n\right)=d^{*}\left(n\right)-w^{H}\left(n\right)z^{*}\left(n\right)$. Then, we search for the optimal solution by stochastic gradient descent method, i.e.,
(15)$$\begin{array}{ccc} w\left(n+1\right) & = & w\left(n\right)-\eta \nabla_{w}\left\{\left[1-\exp\left[-\lambda {\left(e\left(n\right)e^{*}\left(n\right)\right)}^{\alpha }\right]\right]\right\}\\ \; & = & w\left(n\right)+\eta \alpha \lambda \exp\left[-\lambda {\left|e\left(n\right)\right|}^{2\alpha }\right]{\left|e\left(n\right)\right|}^{2\alpha -2}\times \nabla_{w}\left[e\left(n\right)e^{*}\left(n\right)\;\right] \end{array}$$
where the option of the kernel bandwidth alpha is according to *property 3*, so that the cost function is convex and the result of the stochastic gradient descent method is the global optimal solution rather than the local optimal one.

The gradients can be computed as: [24]
(16)$$\begin{array}{ccc} \nabla_{w}\left[e\left(n\right)e^{*}\left(n\right)\right] & = & -\left[e\left(n\right)\frac{\partial y^{*}\left(n\right)}{\partial w(n)}+\frac{\partial y(n)}{\partial w(n)}e^{*}\left(n\right)\right]\\ \; & = & -\left[e\left(n\right)\Phi_{w}\left(n\right)+\Psi_{w}\left(n\right)e^{*}\left(n\right)\right] \end{array}$$
where: (17)$$\begin{array}{c} \Phi_{w}\left(n\right)=\left(\frac{\partial y^{*}\left(n\right)}{\partial w_{R}\left(n\right)}+j\frac{\partial y^{*}\left(n\right)}{\partial w_{I}\left(n\right)}\right) \end{array}$$
(18)$$\begin{array}{c} \Psi_{w}\left(n\right)=\left(\frac{\partial y(n)}{\partial w_{R}\left(n\right)}+j\frac{\partial y(n)}{\partial w_{I}\left(n\right)}\right) \end{array}$$

The gradient vectors (17) and (18) can be written as:(19)$$\Phi_{w}\left(n\right)=\left[\Phi_{a_{1}}\left(n\right),\ldots ,\Phi_{a_{M}}\left(n\right),\Phi_{g_{1}}\left(n\right),\ldots ,\Phi_{g_{M}}\left(n\right),\right.{\left.\;\Phi_{b_{0}}\left(n\right),\ldots ,\Phi_{b_{N}}\left(n\right),\Phi_{h_{0}}\left(n\right),\ldots ,\Phi_{h_{N}}\left(n\right)\right]}^{T}$$
(20)$$\Psi_{w}\left(n\right)=\left[\Psi_{a_{1}}\left(n\right),\ldots ,\Psi_{a_{M}}\left(n\right),\Psi_{g_{1}}\left(n\right),\ldots ,\Psi_{g_{M}}\left(n\right),\right.{\left.\;\Psi_{b_{0}}\left(n\right),\ldots ,\Psi_{b_{N}}\left(n\right),\Psi_{h_{0}}\left(n\right),\ldots ,\Psi_{h_{N}}\left(n\right)\right]}^{T}$$
where R and I are the real and the imaginary part of complex quantities respectively, and $j\;=\;\sqrt{-1}$. To calculate the gradient in (16), items in (17), (18) must be calculated separately, such as:(21)$$\begin{array}{c} \frac{\partial y^{*}\left(n\right)}{\partial a_{m_{R}}\left(n\right)}=y^{*}\left(n-m\right)+\sum_{l=1}^{M}a_{l}^{*}\left(n\right)\frac{\partial y^{*}\left(n-l\right)}{\partial a_{m_{R}}\left(n\right)}+\sum_{l=1}^{M}g_{l}^{*}\left(n\right)\frac{\partial y\left(n-l\right)}{\partial a_{m_{R}}\left(n\right)} \end{array}$$
(22)$$\begin{array}{c} \frac{\partial y^{*}\left(n\right)}{\partial a_{m_{I}}\left(n\right)}=-jy^{*}\left(n-m\right)+\sum_{l=1}^{M}a_{l}^{*}\left(n\right)\frac{\partial y^{*}\left(n-l\right)}{\partial a_{m_{I}}\left(n\right)}+\sum_{l=1}^{M}g_{l}^{*}\left(n\right)\frac{\partial y\left(n-l\right)}{\partial a_{m_{I}}\left(n\right)} \end{array}$$

The feedback in the IIR system leads to the recursions on the right side of (21) and (22). These are the derivatives of the past values to present weights, which are impossible to compute. To avoid this problem, for a small step-size, we can approximate that: (23)$$w\left(n\right)\approx w\left(n-1\right)\approx \cdots \approx w\left(n-\tau \right)\;\tau =\max\{M,N+1\}$$

Thus, the gradient $\Phi_{w}\left(n\right)$ can be written in the following forms,
(24)$$\Phi_{a_{m}}\left(n\right)=y^{*}\left(n-m\right)+\sum_{l=1}^{M}a_{l}^{*}\left(n\right)\Phi_{a_{m}}\left(n-l\right)+\sum_{l=1}^{M}g_{l}^{*}\left(n\right)\Psi_{a_{m}}\left(n-l\right)$$
(25)$$\Phi_{b_{m}}\left(n\right)=x^{*}\left(n-m\right)+\sum_{l=1}^{M}a_{l}^{*}\left(n\right)\Phi_{b_{m}}\left(n-l\right)+\sum_{l=1}^{M}g_{l}^{*}\left(n\right)\Psi_{b_{m}}\left(n-l\right)$$
(26)$$\Phi_{g_{m}}\left(n\right)=y\left(n-m\right)+\sum_{l=1}^{M}a_{l}^{*}\left(n\right)\Phi_{g_{m}}\left(n-l\right)+\sum_{l=1}^{M}g_{l}^{*}\left(n\right)\Psi_{g_{m}}\left(n-l\right)$$
(27)$$\Phi_{h_{m}}\left(n\right)=x\left(n-m\right)+\sum_{l=1}^{M}a_{l}^{*}\left(n\right)\Phi_{h_{m}}\left(n-l\right)+\sum_{l=1}^{M}g_{l}^{*}\left(n\right)\Psi_{h_{m}}\left(n-l\right)$$
and for the gradient vector $\Psi_{w}\left(n\right)$ in (18), similarly, we have,
(28)$$\Psi_{a_{m}}\left(n\right)=\sum_{l=1}^{M}a_{l}\left(n\right)\Psi_{a_{m}}\left(n-l\right)+\sum_{l=1}^{M}g_{l}\left(n\right)\Phi_{a_{m}}\left(n-l\right)$$
(29)$$\Psi_{b_{m}}\left(n\right)=\sum_{l=1}^{M}a_{l}\left(n\right)\Psi_{b_{m}}\left(n-l\right)+\sum_{l=1}^{M}g_{l}\left(n\right)\Phi_{b_{m}}\left(n-l\right)$$
(30)$$\Psi_{g_{m}}\left(n\right)=\sum_{l=1}^{M}a_{l}\left(n\right)\Psi_{g_{m}}\left(n-l\right)+\sum_{l=1}^{M}g_{l}\left(n\right)\Phi_{g_{m}}\left(n-l\right)$$
(31)$$\Psi_{h_{m}}\left(n\right)=\sum_{l=1}^{M}a_{l}\left(n\right)\Psi_{h_{m}}\left(n-l\right)+\sum_{l=1}^{M}g_{l}\left(n\right)\Phi_{h_{m}}\left(n-l\right)$$

So the GMCCC-AIIR can be expressed in the form as:(32)$$\begin{array}{cc} w\left(n+1\right)= & \;w\left(n\right)-\mu \alpha \lambda \exp\left[-\lambda {\left|e\left(n\right)\right|}^{2\alpha }\right]{\left|e\left(n\right)\right|}^{2\alpha -2}\left[e\left(n\right)\Phi_{w}\left(n\right)+\Psi_{w}\left(n\right)e^{*}\left(n\right)\right]\\ = & \;w\left(n\right)+\mu \exp\left[-\lambda {\left|e\left(n\right)\right|}^{2\alpha }\right]{\left|e\left(n\right)\right|}^{2\alpha -2}\left[e\left(n\right)\Phi_{w}\left(n\right)+\Psi_{w}\left(n\right)e^{*}\left(n\right)\right] \end{array}$$

### 3.3. GMCCC-AIIR as a Generalization of ACA-IIR and ACLMS

When $\lambda \to 0^{+}$ and α=1 degenerate to:(33)$$\begin{array}{c} w\left(n+1\right)=w\left(n\right)+\mu \left[e\left(n\right)\Phi_{w}\left(n\right)+\Psi_{w}\left(n\right)e^{*}\left(n\right)\right] \end{array}$$
i.e., the classical ACA-IIR algorithm. On this basis, when feedback within the GMCCC-AIIR is cancelled, that is, the partial derivatives on the right-hand side of (25), (27), (29) and (31) vanish for the widely linear FIR filter, yielding: (34)$$\begin{array}{c} \Phi_{b_{m}}\left(n\right)=x^{*}\left(n-m\right)\;\;\;\;\;\;\;m\;=\;0,...,N \end{array}$$
(35)$$\begin{array}{c} \Phi_{h_{m}}\left(n\right)=x\left(n-m\right)\;\;\;\;\;\;\;m\;=\;0,...,N \end{array}$$
(36)$$\begin{array}{c} \Psi_{b_{m}}\left(n\right)\;=\;\Psi_{g_{m}}\left(n\right)\;=\;0\;\;\;\;\;\;m\;=\;0,...,N \end{array}$$

As desired, the GMCCC-AIIR algorithm (32) now simplifies into the ACLMS algorithm for FIR filters, given by [22]: (37)w(n+1)=w(n)−μe(n)x(n)

#### Reduce the Computational Complexity of AGMCCC-IIR

The weight update of AGMCCC-IIR has a large amount of calculation, and it requires 4×(M+N+1) recursions for the sensitivities $\Phi_{w}\left(n\right)$ and $\Psi_{w}\left(n\right)$. However, this can be simplified to updating only eight sensitivities by the approximation (23). For example,
(38)$$\begin{array}{c} \Phi_{a}\left(n\right)={\left[\Phi_{a_{1}}\left(n\right),\Phi_{a_{2}}\left(n\right),\ldots ,\Phi_{a_{M}}\left(n\right)\right]}^{T} \end{array}$$
(39)$$\begin{array}{c} \Phi_{a_{2}}\left(n\right)=y^{*}\left(n-2\right)+\sum_{l=1}^{M}a_{l}^{*}\left(n\right)\Phi_{a_{2}}\left(n-l\right)+\sum_{l=1}^{M}g_{l}^{*}\left(n\right)\Psi_{a_{2}}\left(n-l\right) \end{array}$$

Further, we define $\Phi_{a}^{F}\left(n\right)$ as follows:(40)$$\begin{array}{c} \Phi_{a}^{F}\left(n\right)={\left[\Phi_{a_{1}}^{F}\left(n\right),\Phi_{a_{2}}^{F}\left(n\right),\ldots ,\Phi_{a_{M}}^{F}\left(n\right)\right]}^{T} \end{array}$$
(41)$$\begin{array}{ccc} \Phi_{a_{2}}^{F}\left(n\right) & = & \Phi_{a_{1}}\left(n-1\right)\\ \; & = & y^{*}\left(n-1-1\right)+\sum_{l=1}^{M}a_{l}^{*}\left(n-1\right)\Phi_{a_{1}}\left(n-l-1\right)\\ & \; & +\sum_{l=1}^{M}g_{l}^{*}\left(n-1\right)\Psi_{a_{2}}\left(n-l-1\right)\\ \; & = & y^{*}\left(n-2\right)+\sum_{l=1}^{M}a_{l}^{*}\left(n-1\right)\Phi_{a_{2}}^{F}\left(n-l\right)+\sum_{l=1}^{M}g_{l}^{*}\left(n-1\right)\Psi_{a_{2}}^{F}\left(n-l\right) \end{array}$$

For a small step-size, $a_{l}^{*}\left(n-1\right)$ and $g_{l}^{*}\left(n-1\right)$ approximate to $a_{l}^{*}\left(n\right)$ and $g_{l}^{*}\left(n\right)$, hence $\Phi_{a_{2}}^{F}\left(n\right)\approx \Phi_{a_{2}}\left(n\right)$. $\Phi_{a}\left(n\right)$ can be approximated, i.e.,
(42)$$\begin{array}{ccc} \Phi_{a}\left(n\right) & = & {\left[\Phi_{a_{1}}^{F}\left(n\right),\Phi_{a_{2}}^{F}\left(n\right),\Phi_{a_{3}}^{F}\left(n\right)\ldots ,\Phi_{a_{M}}^{F}\left(n\right)\right]}^{T}\\ \; & = & {\left[\Phi_{a_{1}}\left(n\right),\Phi_{a_{1}}\left(n-1\right),\Phi_{a_{2}}\left(n-1\right),\ldots ,\Phi_{a_{M-1}}\left(n-1\right)\right]}^{T}\\ \; & = & {\left[\Phi_{a_{1}}\left(n\right),\Phi_{a_{1}}\left(n-1\right),\Phi_{a_{1}}\left(n-2\right),\ldots ,\Phi_{a_{1}}\left(n-M+1\right)\right]}^{T} \end{array}$$

We only need to update $\Phi_{a_{1}}\left(n\right)$ for the sensitivity $\Phi_{a}^{F}\left(n\right)$. This approximation also applies for all other sensitivities.

## 4. Convergence of AGMCCC-IIR

For convenience, we write the algorithm (32) as: (43)$$\begin{array}{c} w\left(n+1\right)=w\left(n\right)+\mu f\left(e\left(n\right)\right)\left[e\left(n\right)\Phi_{w}\left(n\right)+\Psi_{w}\left(n\right)e^{*}\left(n\right)\right] \end{array}$$

$w_{0}$ is defined as the unknown system parameter, and $\tilde{w}\left(n\right)=w_{0}-w\left(n\right)$. In this way,
(44)$$\begin{array}{c} \tilde{w}\left(n+1\right)=\tilde{w}\left(n\right)-\mu f\left(e\left(n\right)\right)\left[e\left(n\right)\Phi_{w}\left(n\right)+\Psi_{w}\left(n\right)e^{*}\left(n\right)\right] \end{array}$$

Thus,
(45)E||w˜n+1||2=E∥w˜(n)∥2−2μERew˜(n)f(e(n))e(n)Φw(n)+Ψw(n)e*(n)+μ2Ee(n)Φw(n)+Ψw(n)e*(n)2|f(e(n))|2

We know that the step-size μ is a small positive constant. If the system converges when n→∞, we can approximate $E\left\{\left|\right|\tilde{w}\left(n+1\right){\left|\right|}^{2}\right\}\approx E\left\{∥\tilde{w}{\left(n\right)∥}^{2}\right\}$. It can be inferred that:(46)0<μ<2ERew(n)f(e(n))e(n)Φw(n)+Ψw(n)e*(n)Ee(n)Φw(n)+Ψw(n)e*(n)2|f(e(n))|2

## 5. Simulation

In this section, we present simulation results to confirm the theoretical conclusions drawn in previous sections. We demonstrate the superiority of the GMCCC-AIIR algorithm compared with the ACA-IIR algorithm in non-Gaussian noise. All the system parameters, signal noise, and input signals are complex valued. The unknown augmented IIR system is given by:(47)$$\begin{array}{ccc} y\left(0\right) & = & 0\\ y\left(n\right) & = & 0.125jy\left(n-1\right)+0.25jy\left(n-2\right)+\left(-0.3+0.7j\right)x\left(n\right)+\left(0.5-0.8j\right)x\left(n-1\right)\\ & \; & +\left(0.2+0.5j\right)x\left(n-2\right)+0.25jy^{*}\left(n-1\right)+0.21jy^{*}\left(n-2\right)+\left(0.32+0.21j\right)x^{*}\left(n\right)\\ & \; & +\left(-0.3+0.7j\right)x^{*}\left(n-1\right)+\left(0.5-0.8j\right)x^{*}\left(n-2\right) \end{array}$$

The real part and the imaginary part of the input signal *x* are Gaussian distributed, with zero mean and unit variance. One hundred Monte Carlo simulations were ran. To evaluate estimation accuracy, the mean square deviation (MSD) is defined by $\mathrm{MSD}={\left|\right|w_{0}-w\left(n\right)\left|\right|}^{2}$.

### 5.1. Complex Non-Gaussian Noise Models

Unlike Gaussian noise, non-Gaussian noise is a random process which the probability distribution function (pdf) of non-Gaussian noise does not satisfy the normal distribution. Generally speaking, the non-Gaussian noise distributions can be divided into two categories: light-tailed (e.g., binary, uniform, etc.) and heavy-tailed (e.g., Cauchy, mixed Gaussian, alphastable, etc.). In the following experiments, four common non-Gaussian noise models, including Cauchy noise, mixed Gaussian noise, alpha-stable noise, and Student’s t noise, are selected for performance evaluation, and the additive complex noise can be written as: $v=v_{re}+jv_{im}$, where $v_{re}$ and $v_{im}$ are obedient to different distributions in different non-Gaussian noise models. The descriptions of these non-Gaussian noise are the following.

#### 5.1.1. Mixed Gaussian Noise

The mixed Gaussian noise model is given by [25]: (48)$$\begin{array}{c} \left(1-\theta \right)N\left(\lambda_{1},v_{1}^{2}\right)+\theta N\left(\lambda_{2},v_{2}^{2}\right) \end{array}$$
where $N\left(\lambda_{i},v_{i}^{2}\right)\left(i=1,2\right)$ denotes the Gaussian distributions with mean values $\lambda_{i}$ and variances $v_{i}^{2}$, and θ is the mixture parameter. Usually, one can set θ to a small value and $v_{2}^{2}\gg v_{1}^{2}$ to represent the impulsive noise. Thus, we define the mixed Gaussian noise parameter vector as $V_{\mathrm{mix}\;}=\left(\lambda_{1},\lambda_{2},v_{1}^{2},v_{2}^{2},\theta \right)$.

#### 5.1.2. Alpha-Stable Noise

The alpha-stable distribution is often used to model the probability distribution of heavy-tailed impulse noise. It is a more generalized Gaussian distribution, or Gaussian distribution is a special case of alpha-stable distribution. It is compatible with many signals in practice, such as noise in telephone lines, atmospheric noise, and backscattering echos in radar systems; even the modeling of economic time series is very successful. The characteristic function of the alpha-stable noise is defined as [26,27]:(49)$$\begin{array}{c} \psi \left(t\right)=\exp\left\{j\delta t-{\gamma |t|}^{\alpha }\left[1+j\beta \mathrm{sgn}\left(t\right)S\left(t,\alpha \right)\right]\right\}\;\left(t\neq 0\right) \end{array}$$
in which:(50)$$\begin{array}{c} S\left(t,\alpha \right)=\left\{\begin{array}{cc} \tan(\alpha \pi /2) & \;\mathrm{if}\;\alpha \neq 1\\ 2/\pi \log|t| & \;\mathrm{if}\;\alpha =1 \end{array}\right. \end{array}$$

From (49) and (50), one can observe that a stable distribution is completely determined by four parameters: (1) the characteristic factor α; (2) the symmetry parameter β; (3) the dispersion parameter γ; and (4) the location parameter δ. Both $v_{re}$ and $v_{im}$ obey the alpha-stable distribution, so we define the alpha-stable noise parameter vector as $V_{\mathrm{alpha}\;}=\left(\alpha ,\;\beta ,\;\gamma \;,\;\delta \right)$.

It is worth mentioning that, in the case of α=2, the alpha-stable distribution coincides with the Gaussian distribution, while α=1,δ=0 is the same as the Cauchy distribution.

#### 5.1.3. Cauchy Noise

The PDF of the Cauchy noise is [28]: (51)$$\begin{array}{c} p\left(v\right)=\frac{1}{\pi \left(1+v^{2}\right)} \end{array}$$

#### 5.1.4. Student’s T Noise

The PDF of the Student’s t noise is [29]: (52)$$\begin{array}{c} p\left(v,n\right)=\frac{\Gamma \left(\frac{n+1}{2}\right)}{\sqrt{n\pi }\Gamma \left(\frac{n}{2}\right)}{\left(1+\frac{v^{2}}{n}\right)}^{\frac{n+1}{2}},-\infty <v<+\infty \end{array}$$
where *n* is the degree of freedom, Γ(·) denotes the Gamma function.

### 5.2. Augmented Linear System Identification

Figure 1 is the block diagram of the system identification, and the length of the adaptive filter is equal to the unknown system impulse response.

First, we demonstrate how kernel bandwidth α affects the convergence performance of GMCCC-AIIR. Figure 2 shows the convergence curves of GMCCC-AIIR with different α, in which the noise chooses mixed Gaussian noise and μ=0.013,λ=0.3. Obviously, the choice of kernel bandwidth has a significant effect on the convergence. In this example, the convergence performance and convergence speed of the proposed algorithm get better when α decreases. Generally speaking, small bandwidth is more robust to impulse noise without considering the convergence rate, and the performance of the algorithm is optimal when α=1.

Second, the stability of GMCCC-AIIR at different step sizes is investigated. Figure 3 shows the convergence performance for different step sizes. The noise is still the mixed Gaussian noise and α=1,λ=0.3. The simulation results show that when the step size is large, such as μ=0.025, the convergence performance gets worse, and the GMCCC-AIIR will diverge if step size continues to increase, which confirms the correctness of the stability analysis in Section 4.

Third, we introduce how the parameter λ will affect the performance of the algorithm. Figure 3 shows the learning curve of GMCCC-AIIR with different λ. The noise is mixed Gaussian noise and α=1,μ=0.013. We can see from Figure 4 that when the parameter λ increases, the convergence speed will slow down. However, when λ is too small, the GMCCC-AIIR algorithm is approximate to the ACA-IIR, and the convergence performance is poor in the non-Gaussian noise model. Therefore, we should choose the appropriate parameter λ according to different situations.

Fourth, we introduce how pairwise parameters α, μ, and λ affect the steady-state excess MSD (EMSD) and give 3D diagrams of EMSD. The number of iterations is increased to 15,000 to ensure the convergence of the algorithm, and the additive noise is still mixed Gaussian noise. EMSD equals the average MSD of the last 1000 iterations. Figure 5 shows that EMSD of GMCCC-AIIR mainly depends on α and μ. The performance of the algorithm get worse when μ increases and α approaches 1, the algorithm performs best. Combined with Figure 4 and Figure 5b,c, λ mainly affects the convergence speed of the GMCCC. When λ approaches 0, the robustness of the algorithm to non-Gaussian noise gets worse, the outliers of EMSD increase, and the algorithm may even diverge.

Fifth, we compare the performance of GMCCC-AIIR and ACA-IIR under four noise distributions. In the simulation, the mixed Gaussian noise and alpha-stable noise parameters are set separately at $V_{\mathrm{mix}\;}=\left(0,0,0.01,100,0.03\right)$ and $V_{\mathrm{alpha}\;}=\left(1.4,\;0,\;0.3\;,\;0\right)$, the freedom parameter of student noise n is set to 2, and the Cauchy noise is the standard form. The step-sizes are chosen such that both algorithms have almost the same initial convergence speed. The simulation results are shown in Figure 6. As expected, the proposed GMCCC-AIIR algorithm can achieve better steady-state performance than ACA-IIR significantly in these non-Gaussian noise models. The ACA-IIR algorithm diverges after encountering impulse and the MSD approaches infinity at this time. The convergence process cannot be well observed when trying to display the ACA-IIR and the GMCCC-AIIR learning curve in the same coordinate system. Therefore, we limit the height of all simulation results.

## 6. Conclusions

In this paper, we propose an adaptive algorithm for augmented IIR filter based on generalized maximum complex correlation entropy criterion. We study the convergence performance, providing the bound for the step size. Moreover, computational complexity is reduced by making use of the redundancy in the state vector of the filter. We also prove that ACA-IIR and ACLMS are special cases of GMCCC-AIIR. The simulation results verify the theoretical conclusion and show how parameters affect the convergence performance of GMCCC-AIIR and superiority of the GMCCC-IIR algorithm compared with the MSE-based algorithm ACA-IIR when the noise is non-Gaussian distribution.

## Figures and Tables

**Figure 1 entropy-24-01008-f001:**
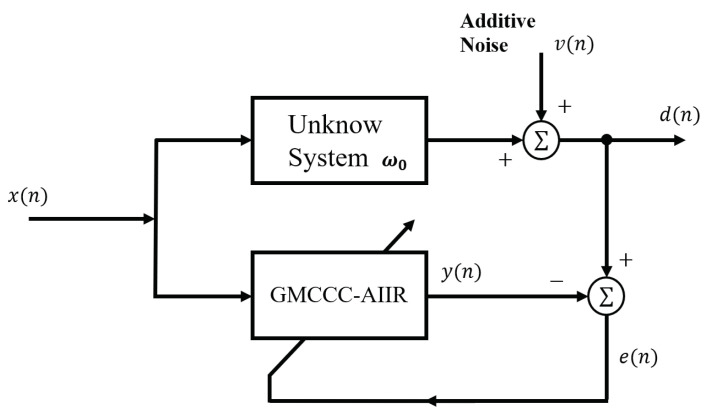
System Identification Configuration.

**Figure 2 entropy-24-01008-f002:**
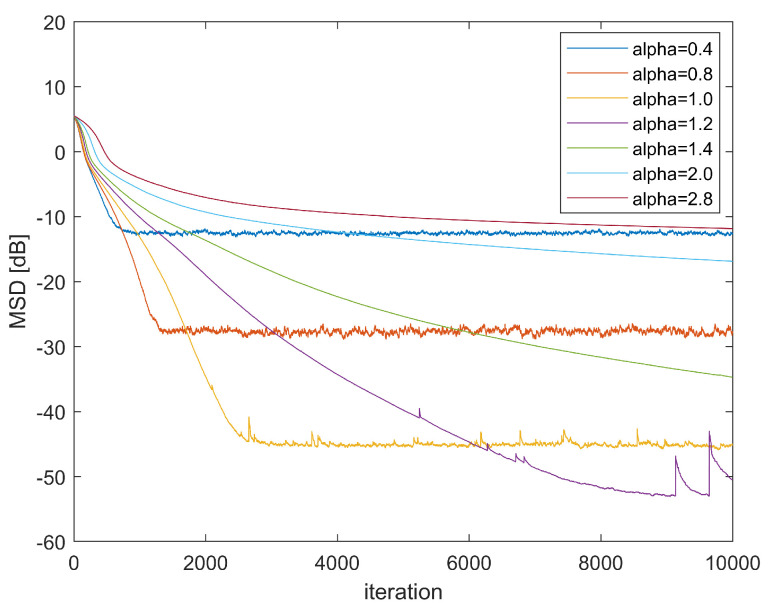
Learning Curve in different α.

**Figure 3 entropy-24-01008-f003:**
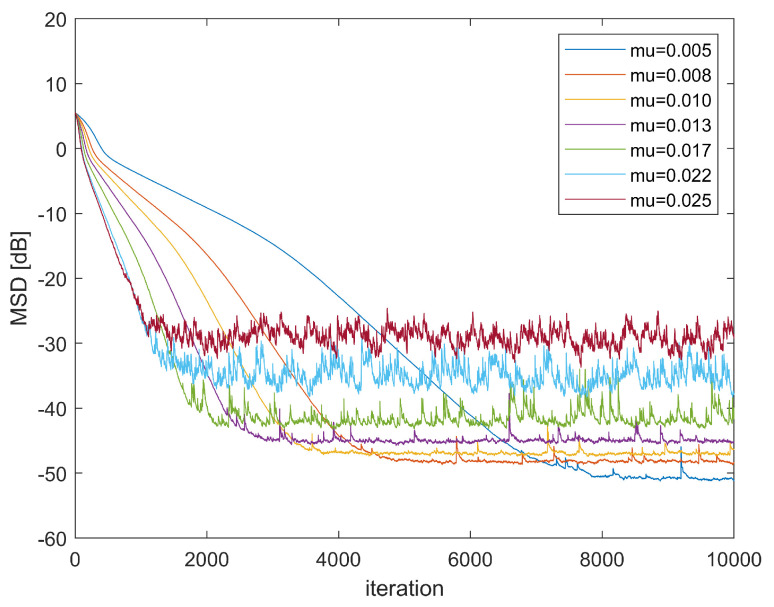
Learning Curve in different μ.

**Figure 4 entropy-24-01008-f004:**
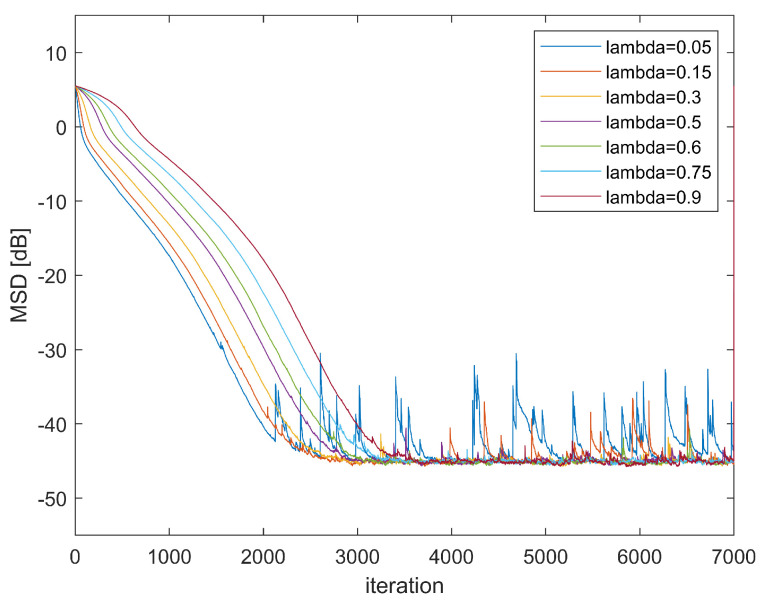
Learning Curve in different λ.

**Figure 5 entropy-24-01008-f005:**
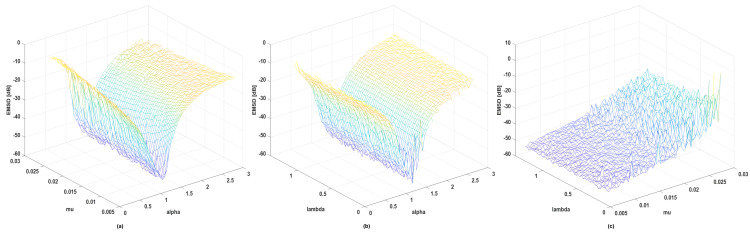
EMSD with different pairwise parameters (**a**) α and μ (λ=0.3) (**b**) α and λ (μ=0.013) (**c**) μ and λ (α=1).

**Figure 6 entropy-24-01008-f006:**
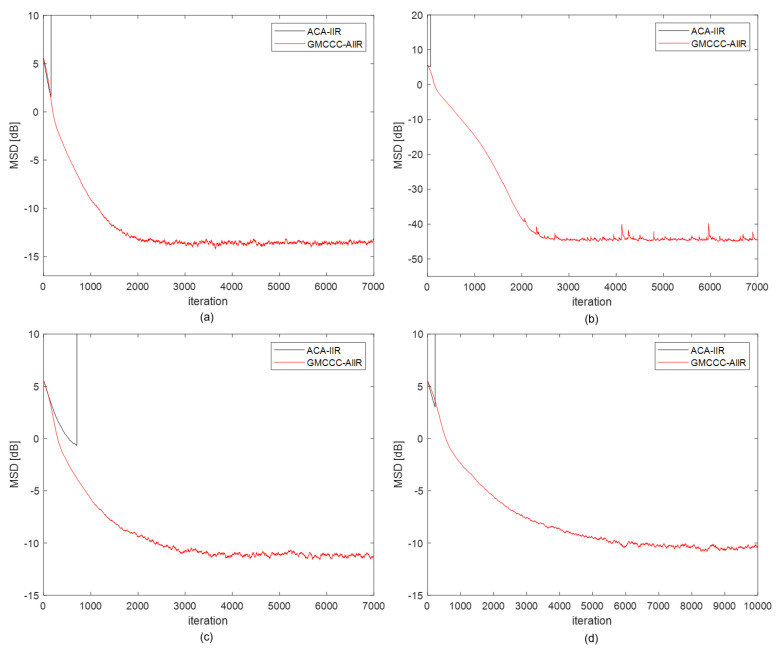
Learning Curve with different noise: (**a**) alpha-stable noise, (**b**) mixed Gaussian noise, (**c**) Cauchy noise, (**d**) Student’s t noise.

## Data Availability

Not applicable.

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
