# Peer review of "Generalized Maximum Complex Correntropy Augmented Adaptive IIR Filtering"

_entropy, 2022, doi:10.3390/e24071008_

Round 1
Reviewer 1 Report
Please, kindly refer to comments attached in the PDF file

Reviewer 2 Report
This contribution presents original ideas in the study and advances the previous research in this area. The paper is very good and deserves publication. The level of the originality of contribution to the existing knowledge with an emphasis on the paper’s innovativeness in both theory development and methodology used in the study is very high.
This work makes a significant practical contribution and it makes impact on the research work on the research community.
The quality of arguments, the critical analysis of concepts, theories and findings, and consistency and coherency of debate are well addressed in this paper.
The paper has a good writing style in term of accuracy, clarity, readability, organization, and formattingNevertheless, the following issues should be addressed before publishing the paper.
- In the definition of the alpha-stable noise (Equation (48) and (49)), plese clarify if for "t=0" we have a singularity.
- Please clarify why you introduced the concept of alpha-stable noise.
- Please emphasize the difference between the case Gaussian noise and non-Gaussian noise.
- Please clarify the practical meaning of the concept of meaning of the Complex Correntropy and the difference between a "normal Entropy". The following literature, which is dedicated to Signal Processing using entropy functions, can help to inspire the writer and the reader to understand the meaning of this question. some papers listed below are published by MDPI Journals.
-Please try to clarify better the improvement of the calculation complexity also trying to address other methods as possible comparison. The listed literature here below can inpire you.Pleas clyrify the
The paper is very good and deserves publication.
Concerning the cited literature you can consider the following paper to improve the tutorial aspects of the paper and to answer some queries listed above.
Mercorelli, P., “Biorthogonal wavelet trees in the classification of embedded signal classes for intelligent sensors using machine learning applications”, 2007. Journal of Franklin Institute (Elsevier Publishing), vol. 344, no. 6, pp. 813-829..
Schimmack, M. et al. An Adaptive Derivative Estimator for Fault-Detection Using a Dynamic System with a Suboptimal Parameter. Algorithms 2019, 12, 101. https://doi.org/10.3390/a12050101
Mercorelli, Paolo, A Fault Detection and Data Reconciliation Algorithm in Technical Processes with the Help of Haar Wavelets Packet, 2017, J Algorithms (MDPI), pp.1999-4893, vol. 10, no. 1, doi:10.3390/a10010013
Mercorelli, P., Denoising and Harmonic Detection Using Nonorthogonal Wavelet Packets in Industrial Applications, J Journal of Systems Science and Complexity, 2007. vol. 20, no. 3, pp. 1559-7067
M Schimmack et al.An on-line orthogonal wavelet denoising algorithm for high-resolution surface scans, 2018. Journal of the Franklin Institute 355 (18), 9245-9270
P. Mercorelli A denoising procedure using wavelet packets for instantaneous detection of pantograph oscillations, 2013.
Mechanical Systems and Signal Processing 35 (1-2), 137-149
Schimmack M. et al. “A Structural Property of the Wavelet Packet Transform Method to Localise Incoherency of a Signal”, 2019. Journal of the Franklin Institute, vol. 356, no. 16, pp. 10123-10137.
Reviewer 3 Report
In the submitted manuscript, the authors propose a new algorithm for the augmented adaptive IIR filter based on generalized maximum complex correntropy criterion.
In general, the article may be of interest to scolars in the field of digital signal processing. However, in order to adapt the the material to a wider range of readers, a major revision is necessary to improve the presentation.
The transition to the discussion of FIR filters in the introduction is too abrupt and needs some preliminary complement.
It is not clear from the text how the properties of generalized complex correntropy affect the derrivation of the cost function for the proposed algorithm.
Please insert the block diagram of the system identification problem and indicate the place of the GMCCC-AIIR algorithm in it.
Along with figures 2-4, please add 3D diagrams of learning performance rate regarding pairwise parameters α, µ, and λ.
Round 2
Reviewer 1 Report
I would like to thank to the authors for taking into consideration all the commets I presented. I see a huge improvement in the quality of the paper.
Author Response
Response to Reviewer 1 Comments
Thanks very much for taking your time to review this manuscript. We really appreciate all your comments and suggestions, which improve the quality of this paper a lot. In the latest version, we haved proofread the manuscript again aim at correcting language stylistic and content formatting. We have also corrected the labeling problems in figures 2, 3, 4 and 6.
Reviewer 3 Report
The authors have correctly addressed all the given comments. However, the manuscript still needs the proofread aimed at correcting language stylistic and content formatting. Particularly, please, pay attention to the absence of axes labels in Figures 2-4.
Author Response
Response to Reviewer3 Comments
Thanks very much for taking your time to review this manuscript. We really appreciate all your comments and suggestions, which improve the quality of this paper a lot.
Point 1: However, the manuscript still needs the proofread aimed at correcting language stylistic and content formatting
Response 1: We haved proofread the manuscript again to edit English language and style required.
Point 2: Particularly, please, pay attention to the absence of axes labels in Figures 2-4.
Response 2: We are sorry we did not notice the problems of figures’ axis label in the manuscript. In the latest version, we have add the axis label in fugure 2-4, and we have corrected the wrong axis label in figure 6.